# Inhibition of Human Neutrophil Functions In Vitro by Multiple Sclerosis Disease-Modifying Therapies

**DOI:** 10.3390/jcm9113542

**Published:** 2020-11-02

**Authors:** Sara Scutera, Tiziana Musso, Paola Cavalla, Giorgia Piersigilli, Rosaria Sparti, Sara Comini, Marco Vercellino, Anna Maria Cuffini, Giuliana Banche, Valeria Allizond

**Affiliations:** 1Department of Public Health and Pediatrics, University of Torino, 10126 Turin, Italy; sara.scutera@unito.it (S.S.); tiziana.musso@unito.it (T.M.); giorgia.piersigilli@unito.it (G.P.); rosaria.sparti@unito.it (R.S.); sara.comini@unito.it (S.C.); giuliana.banche@unito.it (G.B.); valeria.allizond@unito.it (V.A.); 2Department of Neuroscience, University of Torino, 10126 Turin, Italy; paola.cavalla@unito.it (P.C.); marco.vercellino@unito.it (M.V.)

**Keywords:** disease-modifying therapies, multiple sclerosis, neutrophil functions, *Klebsiella pneumoniae*

## Abstract

There is a growing optimism about the potential of new disease-modifying therapies (DMTs) in the management of relapsing-remitting multiple sclerosis (RRMS) patients. However, this initial enthusiasm has been tempered by evidence indicating that multiple sclerosis (MS) patients undergoing DMT may be at higher risk of developing infections through incompletely understood mechanisms. As neutrophils provide the first line of defense against pathogens, here we have compared the effects of some of the commonly used MS DMTs (i.e., moderate-efficacy injective, first-line: interferonβ-1b (IFNβ-1b), glatiramer acetate (GA); and high-efficacy, second-line: fingolimod (FTY) and natalizumab (NAT)) on the in vitro viability and functions of neutrophils isolated from healthy subjects. All the DMTs tested impaired the ability of neutrophils to kill *Klebsiella pneumoniae*, whereas none of them affected the rate of neutrophil apoptosis or CD11b and CD62L cell surface expression. Intriguingly, only FTY exposure negatively affected *K. pneumoniae*-induced production of reactive oxygen species (ROS) in polymorphonuclear leukocytes (PMNs). Furthermore, neutrophils exposed to *K. pneumoniae* secreted enhanced amounts of CXCL8, IL-1β and TNF-α, which were differentially regulated following DMT pretreatment. Altogether, these findings suggest that DMTs may increase the susceptibility of MS patients to microbial infections, in part, through inhibition of neutrophil functions. In light of these data, we recommend that the design of personalized therapies for RRMS patients should take into account not just the mechanism of action of the chosen DMT but also the potential risk of infection associated with the administration of such therapeutic compounds to this highly vulnerable population.

## 1. Introduction

Over the past decade, an increasing number of disease-modifying therapies (DMTs) have been successfully launched on the market for the treatment of relapsing-remitting multiple sclerosis (RRMS) patients [1,2,3,4]. DMTs can indeed reduce to various extents, or even halt, MS progression, de facto reducing the likelihood of lesion development or enlargement, clinical relapses and stepwise disability accumulation [4,5].

Even though DMTs act through different mechanisms of action, they all appear to interfere with the immune response of the patient [2]. In particular, currently available DMTs, such as fingolimod (FTY) and natalizumab (NAT), inhibit the trafficking of immune cells with different mechanisms, while alemtuzumab, ocrelizumab, rituximab and cladribine appear to promote the depletion of specific sub-populations of immune cells. In addition, other DMTs such as glatiramer acetate (GA), interferons (IFNs) and dimethyl fumarate have been shown to affect immune cell functions or, in the case of teriflunomide, cell replication [6,7].

Two treatment approaches have emerged in MS treatment [8,9]. The most common approach is based on first-line use of moderate-efficacy DMTs, which have generally good safety profiles, with escalation to high-efficacy DMTs only in the presence of breakthrough disease activity (i.e., relapses or new lesions in Magnetic Resonance Imaging (MRI)). High-efficacy second-line DMTs may carry higher risks than moderate-efficacy DMTs [10]: occurrence of progressive multifocal leukoencephalopathy (PML), infections and secondly acquired autoimmunity. Therefore, use of high-efficacy DMTs is post-posed and reserved for patients experiencing inadequate suppression of disease activity on a moderate-efficacy DMT. The alternative approach is based on the use of high-efficacy therapies from the onset, implying earlier and higher exposure to potential risks. Usually this approach is reserved to the most active forms of MS [8,9].

Despite the effectiveness of DMTs in treating RRMS patients, it has become increasingly clear that their use, in particular second-line DMTs, may be associated with increased risk of microbial infections, both opportunistic and community-acquired [1,2,3,11,12,13,14], now representing one of the main issues to be dealt with when choosing the most appropriate therapeutic course of action [13]. This is particularly relevant in light of the fact that DMTs, besides inhibiting normal T cell, and monocyte and neutrophil functions, can weaken humoral immunity, thereby increasing the risk of bacterial, viral, fungal or parasitic infections [6].

Polymorphonuclear leukocytes (PMNs) play a key role in the elimination of pathogens from the body through a wide range of effector functions [15,16,17,18,19,20]. As we have previously shown that PMNs isolated from RRMS patients display reduced intracellular killing activity [20], here we sought to determine whether treatment with first-line (moderate-efficacy) DMTs, such as IFNβ-1b (Extavia^®^) and GA (Copaxone^®^), or second-line (high-efficacy) DMTs, such as FTY (Gilenya^®^) and NAT (Tysabri^®^), would affect neutrophil functions in vitro.

Our results indicate that DMTs impair the ability of PMNs from healthy subjects (HSs) to kill *Klebsiella pneumoniae* and to secrete pro-inflammatory cytokines, suggesting that a direct effect of these agents on neutrophil functions may also weaken the immune response against pathogens in MS patients.

## 2. Experimental Section

### 2.1. PMN Isolation and Purification

As previously detailed [20,21,22], PMNs were isolated from HS peripheral venous blood by gravity in 2.5% dextran (500,000 molecular weight) in standard saline solution (ratio 1:1), and subsequently, by Ficoll-Paque density gradient centrifugation (twice at 1200× *g* for 15 min). After an erythrocyte hypotonic shock, pure PMNs were obtained and adjusted to 10^6^ cells/mL in RPMI 1640 upon Bürker chamber counting (Marienfield, Germany).

### 2.2. DMT Drugs

The first-line DMT IFNβ-1b (Extavia^®^), used at 250 µg/mL, was purchased from Novartis Europharm Limited (Horsham, UK); GA (Copaxone^®^), a mixture of synthetic polypeptides composed of four amino acids used at 15 µg/mL, was purchased from Teva Pharmaceuticals Ltd. (Castleford West Yorkshire, UK). The second-line option NAT (Tysabri^®^), a highly specific α4-integrin antagonist used at 25 µg/mL, was acquired from Biogen Idec Limited (Maidenhead, UK). The other second-line agent FTY (Gilenya^®^), a sphingosine 1-phosphate analogous used at 10 µM, was obtained from Novartis Europharm Limited (Horsham, UK). These DMT concentrations were chosen among those used in other in vitro experimental assays or those achieved in vivo [23,24,25,26,27].

### 2.3. Evaluation of Neutrophil Intracellular Antimicrobial Activity

The intracellular bacterial killing activity of DMT-pretreated PMNs was investigated using a clinical strain of *K. pneumoniae*, a pathogen commonly found to be responsible for frequent infections in MS patients. Briefly, PMNs (10^6^ cells/mL) were incubated for 60 min with the specific DMT, used at the concentrations mentioned above, then bacteria (10^7^ CFU/mL) were added at 37 °C in a shaking water bath for 30 min to allow phagocytosis to proceed, and subsequently the PMN-bacterium mixtures were centrifuged at 1200× *g* for 5 min and washed with phosphate saline to remove the free extracellular bacteria. An aliquot of the cells containing bacteria was taken, and was lysed by adding sterile water, and a viable count of intracellular *Klebsiellae* was performed (T0). The cells were then incubated further, and at intervals (T30, T60 and T90), the viable counts of the surviving intracellular bacteria were measured in the same way. The PMN killing values were expressed as the survival index (SI), which was calculated by adding the number of surviving microorganisms at T0 to the number of survivors at Tx (T30, T60 and T90) and dividing by the number of survivors at T0. According to this formula, if bacterial killing was 100% effective, the SI would be 1 [20,21,22].

### 2.4. Oxidative Burst Assay

To asses neutrophil reactive oxygen species (ROS) production, cells were pretreated for 60 min with the indicated DMT and then incubated in polypropylene tubes with 5 µM PMA (Sigma Aldrich, St. Louis, MO, USA), *K. pneumoniae* (10:1), or left untreated, at 37 °C for 30 min. Successively, DHR123 (Sigma Aldrich, St. Louis, MO, USA) at a concentration of 30 µg/mL was added to the stimulated and resting samples. After 5 min at 37 °C, cells were washed twice in 1× phosphate buffered solution (PBS), resuspended in 1 mL of 1% paraformaldehyde to stabilize the cells, and acquired using FACSCalibur (BD Biosciences, San Jose, CA, USA). Analysis was performed by FlowLogic software (Miltenyi Biotech, Bergisch Gladbach, Germany).

### 2.5. Analysis of PMN Apoptosis and Surface Markers by Flow Cytometry

Purified PMNs were harvested in polypropylene tubes at a concentration of 1 × 10^6^/mL and stimulated or not with the specific DMT for 3 h at 37 °C in RPMI 10% FCS. Cells were subsequently washed, resuspended in Annexin V Binding Buffer, and stained with FITC-annexin V and propidium iodide (PI) (Biolegend, San Diego, CA, USA), as indicated by the manufacturer’s instructions, to measure apoptosis and necrosis, respectively. Flow cytometric analysis was performed using FlowLogic software. For some experiments, cells were pretreated for 60 min with the specific DMT and then incubated for 2 h with *K. pneumoniae* (10:1), or left untreated before proceeding with the annexin V/PI staining.

To evaluate changes in PMN surface markers, cells were pretreated for 60 min with IFNβ-1b, as first-line drug, or NAT, as second-line drug, and then incubated with *K. pneumoniae* (10:1), or left untreated, at 37 °C for 30 min. Cells were then washed in FACS buffer (PBS, 2% fetal calf serum, 2 mM EDTA) and stained using anti-human CD11b APC, anti-human CD62L FITC or specific isotype control antibodies (mouse IgG2a APC and mouse IgG1 FITC, respectively) (all from BD Biosciences, San Jose, CA, USA). Dead cells were excluded by PI staining (Sigma Aldrich, St. Louis, MO, USA). Cells were gated on forward and side scatter to exclude debris and cell aggregates and analyzed by BD FACSCalibur (BD Biosciences, San Jose, CA, USA) using FlowLogic software (Miltenyi Biotech, Bergisch Gladbach, Germany). Results were recorded as geometric mean fluorescence intensity (gMFI), which represents the cell surface receptor density.

### 2.6. Quantification of Cytokine Production

The culture supernatants of pretreated PMNs, exposed to *K. pneumoniae* (ratio 10:1) for T30, T60, T90 and T180, were analyzed by ELISA (R&D Systems, Minneapolis, MN, USA) to determine the production of the pro-inflammatory cytokine IL-1β, CXCL8 and TNF-α according to the manufacturer’s instructions.

### 2.7. Data Analysis

Differences between DMTs pretreated and untreated HS PMNs were calculated by Mann–Whitney test or by unpaired T-test, as appropriate. Statistical analyses were all carried out using the GraphPad Prism version 8 for Windows (GraphPad Software, San Diego, CA, USA). The results were analyzed by descriptive statistics (average values ± SEM or SD); *p* < 0.05 was considered significant.

## 3. Results

### 3.1. Direct Effect of DMTs on PMN Intracellular Killing Activity of Klebsiella pneumoniae In vitro

Our primary objective was to evaluate the direct effect of DMTs on the intracellular killing activity of PMNs isolated from healthy subjects (HSs), hereinafter simply referred to as PMNs. For this purpose, PMNs were pre-incubated for 60 min with different first-line (i.e., IFNβ-1b and GA) or second-line (i.e., NAT and FTY) agents and subsequently tested for their intracellular killing activity against *K. pneumoniae*, measured as survival index (SI) and as the corresponding killing percentage. DMT-treated PMNs showed a statistically significant reduction of intracellular bacterial killing compared to control PMNs throughout the observation period (Figure 1 and Appendix A). In particular, untreated PMNs were able to kill *K. pneumoniae* with values of 49% (SI = 1.51), 26% (SI = 1.74) and 0% (SI > 2) at T30, T60 and T90, respectively. In contrast, treatment of PMNs with IFNβ-1b and GA severely impaired neutrophil-mediated bacterial killing. Of note, the moderate killing activity observed in IFNβ-1b- and GA-treated PMNs at T30 (13%, SI = 1.87 and 23%, SI = 1.77, respectively) was completely inhibited at T60 and T90. Furthermore, NAT-treated PMNs killed intracellular *K. pneumoniae* to a similar extent to what was observed for IFNβ-1b and GA, with values ranging from 31% to 0% within 90 min of incubation, whereas with FTY, no PMN killing activity was recorded throughout the whole incubation time.

### 3.2. Effect of DMTs on Apoptosis

To investigate the effects of DMTs on apoptosis, PMNs were incubated with the indicated DMT and then analyzed by flow cytometry upon double-staining with annexin V and PI. This approach allows the distinction of following three subgroups of cells: (1) annexin V-negative and PI-negative viable cells; (2) annexin V-positive and PI-negative early apoptotic cells with undamaged membranes; and (3) annexin V-positive and PI-positive late apoptotic/necrotic cells with disrupted plasma membranes. As shown in Figure 2A (and Appendix A), all drugs tested did not significantly affect the constitutive rate of apoptosis in neutrophils. Conversely, PMNs incubated with *K. pneumoniae* displayed decreased cell viability, as attested by the presence of more late apoptotic cells compared to unstimulated control cells. Lastly, surface expression of phosphatidylserine, the marker of annexin V, and PI positivity remained unchanged in DMT-pretreated neutrophils (Figure 2B).

### 3.3. Effect of DMTs on ROS Production in Resting versus Stimulated PMNs

Next, to determine whether DMTs affected the respiratory burst, we assessed the oxidation levels of the dihydrorhodamine 123 (DHR123) compound by flow cytometry in resting or *K. pneumoniae* or phorbol myristate acetate (PMA)-activated PMNs. While we observed almost negligible levels of ROS production in control or DMT-treated PMNs, stimulation with *K. pneumoniae* or PMA was accompanied by a substantial increase in the percentage of bursting neutrophils (Figure 3) as well as ROS production (measured as gMFI). Interestingly, the respiratory burst response was not affected by IFNβ-1b, GA and NAT treatment, while FTY significantly reduced ROS production (Figure 3, lower right panel) in bacteria- or PMA-stimulated cells, as judged by gMFI measurement (370.3 ± 47.7 in PMA-stimulated cells vs. 239.6 ± 21.7 in FTY + PMA stimulated cells).

### 3.4. Effect of DMTs on the Expression of the Neutrophil Adhesion Molecules CD11b and CD62L

PMN activation is generally accompanied by changes in the expression profiles of cell surface adhesion molecules. Specifically, L-selectin (CD62L) is downregulated through stimulus-induced shedding, whereas β2 integrins (CD11b/CD18) are upregulated following translocation from intracellular granules [28,29]. Thus, the effects of IFNβ-1b and NAT on CD11b and CD62L expression levels were evaluated by FACS. *K. pneumoniae* rapidly upregulated cell surface expression of CD11b on PMNs, and this effect was accompanied by a decline in CD62L expression (Figure 4). Neither IFNβ-1b nor NAT was able to modify its expression profile in resting or activated PMNs.

### 3.5. Cytokine Release Pattern of DMT Pretreated PMNs upon K. pneumoniae Stimulation

In light of the aforementioned cell killing results, we next assessed the expression levels of the pro-inflammatory cytokines CXCL8, IL-1β and TNF-α in the supernatants of DMT-treated PMNs at different time points (from T0 to T180 min) following *K. pneumoniae* exposure. The cytokine concentrations, measured as pg/mL, were expressed as percentages of production relative to untreated PMNs following *K. pneumoniae* stimulation at T30 set as 100% (expressed as mean ± SD: CXCL8 10158 ± 3490 pg/mL, IL-1β 95 ± 47 pg/mL and TNF-α 56 ± 25 pg/mL). The cytokine release kinetics and the degree of statistical significance are depicted in Figure 5 and Figure 6. IFNβ-1b- or FTY-treated PMNs (Figure 5, left panel and Figure 6, right panel, respectively) showed a decrease in CXCL8 secretion, whereas treatment of PMNs with GA or NAT (Figure 5, right panel and Figure 6, left panel, respectively) led to a reduction in IL-1β and TNF-α release.

## 4. Discussion

Thanks to recent advances in our understanding of the immunological mechanisms underlying MS pathogenesis, an increasing number of DMTs have become available to treat MS patients, ranging from non-selective immunosuppressive therapies to targeted immune interventions [11,12,30]. However, the initial enthusiasm about DMT efficacy has been partly tempered by the increased infectious risk [2]. Recurrent microbial urinary and respiratory tract infections have been in fact reported in RRMS patients treated with high-efficacy drugs [1,14]. Consequently, treatment decisions have become more complex as they require exhaustive information on the potential adverse effect of the DMT being used [1,2,13,31].

The present study demonstrates that DMTs, at concentrations similar to those achieved by standard dosing regimens in human subjects, hamper the ability of neutrophils to kill *K. pneumoniae*. FTY was the most effective agent in attenuating PMN intracellular killing as it completely abrogated killing activity at the 30 min time point. A rapid effect was also observed following IFNβ-1b treatment, while NAT and GA exerted their inhibitory effect to a lesser extent.

To better clarify the effect of DMTs on neutrophil activity, we also asked whether DMTs could influence PMN survival. Neutrophils are characterized by a short lifespan, because they undergo constitutive apoptosis, and phagocytosis typically accelerates this process [32]. All tested DMTs failed to modulate both spontaneous and *K. pneumoniae-*induced apoptosis. Furthermore, using DHR123 as fluorochrome, following *K. pneumoniae* or PMA stimulation, we did not record a significant difference in the respiratory burst of neutrophils treated with NAT, IFNβ or GA. By contrast, FTY significantly inhibited superoxide production. Lastly, as neutrophil activation is associated with upregulation of CD11b and shedding of CD62L, we sought to determine the effect of DMTs on the expression of these cell surface molecules. However, none of the aforementioned DMTs affected the expression of CD11b and CD62L in bacteria-stimulated neutrophils.

Overall, the observation that PMNs treated with DMTs display an impaired ability to kill internalized bacteria suggests that depressed functional properties of MS neutrophils, previously demonstrated *ex vivo*, may be due to a direct effect of DMTs on the neutrophils themselves rather than being caused by alteration of the immune/inflammatory environment. Fittingly, we have previously shown that PMNs from RRMS patients treated with first-line (IFNβ and GA) or second-line (FTY and NAT) drugs display impaired microbicidal activity [20]. In good agreement, PMNs and monocytes collected from MS patients subjected to a prolonged IFNβ therapy showed a decrease in both phagocytosis and *Candida albicans* killing activity [33]. With regard to GA, known for its suppressive effect on lymphocytes and monocytes [7,11,12,34], to date no studies are currently available on its action on neutrophil functions. As for NAT, Fleming et al. (2010) [25] reported that exposure to an anti-α4β1 integrin antibody did not alter rat neutrophil phagocytic or oxidative activities. Likewise, despite detecting impaired PMN killing activity, we did not notice any difference in ROS production or surface marker expression levels in these cells after DMT stimulation.

Among the DMTs tested, FTY exerted the most potent inhibitory effect on bacterial killing and ROS production in neutrophils. Under our experimental conditions, FTY-mediated inhibition of respiratory burst activity could not be ascribed to drug-induced apoptosis of PMNs. FTY is an analog of sphingosine-targeting S1P receptors (S1PRs) involved in trafficking and activation of immune cells. Once phosphorylated in vivo by sphingosine kinase, FTY binds S1PR1, 3, 4 and 5, inducing their internalization and degradation, thereby inhibiting S1P downstream signaling [35]. Better known for its ability to prevent the egress of naïve and central memory T-cells from lymph nodes, FTY also affects dendritic cell (DC) migration, modulates DC proinflammatory signaling, and influences the phenotype and function of circulating T cells [36,37]. S1PRs have also been linked to neutrophil migration [38], and FTY is known to alter neutrophil influx to lymph nodes in an inflammation model induced by immunization of BALB/c mice with OVA emulsified in CFA [39]. In this context, Gorlino CV et al. (2014) [39] confirmed a direct effect of FTY on S1PRs in neutrophils and, in line with our results, failed to detect apoptotic cell death in FTY-treated neutrophils in vitro. No significant effect of FTY on apoptosis and ROS production was detected in feline neutrophils as well [40]. Conversely, high doses (20 µM) of FTY have been shown to induce an atypical cell death in human neutrophils, characterized by HSP27 externalization and lack of typical features of apoptosis. According to the literature, the effect of FTY on apoptosis and ROS production may vary depending on the pathological environment, cell type and dose. For example, FTY exhibits antitumor activity by inducing apoptosis and ROS generation [41] in various cancer cells, while it activates anti-apoptotic and anti-oxidative pathways in cardiomyocytes, thus exerting a cardioprotective effect [42]. In our experimental setting, FTY was the only DMT capable of impairing ROS production following stimulation of PMNs with *K. pneumoniae* and PMA. Considering that S1P can enhance neutrophil activation and prime the respiratory burst [43], our results are consistent with an antagonizing effect of FTY on SP1 signaling. In line with this observation, the activated status of neutrophils from asthmatic patients, in term of ROS production in vitro, was reduced by FTY treatment [44].

Our findings also indicate that DMTs affect the ability of PMNs to secrete pro-inflammatory cytokines. In this regard, unbalanced cytokine secretion has been shown to play a role in MS pathogenesis. This is of particular relevance given that DMT use appears to exert beneficial effects in MS patients partly through modulation of cytokine production [45]. In support of this hypothesis, IFNβ administration significantly reduces TNF-α and IFN-γ expression levels and upregulates IL-10 secretion [46].

Previously, serum CXCL8 and its secretion from PBMCs, significantly higher in untreated MS patients compared to HS, were shown to be reduced following IFNβ-1a therapy [46]. In line with these results, we found a significant downregulation of CXCL8 secretion upon *K. pneumoniae* stimulation of IFNβ-treated neutrophils.

Similarly to IFNβ, GA skews the balance from pro-inflammatory (Th1) toward anti-inflammatory (Th2) immune responses [1,13]. Consistently, increased secretion of IL-10 and reduced production of IL-12 from monocytes and DCs were reported following treatment with GA [34,47,48]. This is in line with our results showing that GA-treated PMNs display impaired IL-1β and TNF-α production.

Previous studies also demonstrated a positive association between NAT treatment and reduced cerebrospinal fluid (CSF) and serum concentrations of TNF-α, IL-1β, IL-6 and CXCL8 [49,50]. Consistently, we show that NAT-induced impairment of neutrophil intracellular killing correlates with decreased of IL-1β and TNF-α production. Even though FTY has been found to significantly reduce the production of pro-inflammatory cytokines, such as IFN-γ and IL-17, by CD4^+^ T cell subsets, other studies have failed to detect a direct effect of FTY treatment on the serum levels of CXCL13, IL-23, IL-4, IL-6, VLA-4, CXCL10, TNF-α, IL-13 and IL-22. Interestingly, and in good agreement with our results, FTY suppressed LPS-induced CXCL8 in neutrophils isolated from asthmatic patients [44]. Pro-inflammatory cytokines and chemokines attract and activate leukocytes to phagocytose pathogen and to release toxic substances. In the past, it was demonstrated that IL-1β, INF-γ and TNF-α enhance phagocytosis of bacteria but do not have consistent effect on bacterial killing [51]. Furthermore, recent observations have expanded our understanding of a direct chemokine-mediated bactericidal activity with diverse and complex actions [52]. On this basis, we could not exclude the autocrine actions of these cytokines and the influence of decreased cytokine levels in the impaired killing activity.

To our knowledge, the present study is the first comparative evaluation of the effects of DMTs on neutrophil functions regulating the host immune response. From our data, it is tempting to speculate that the reduction in neutrophil killing activity observed following DMT treatment may be directly responsible for impaired antibacterial immunity in MS patients after prolonged exposure to these drugs. Furthermore, our findings together with our previous work [20] suggest that impaired PMN functions in DMT-treated MS patients could contribute to the increased risk of microbial infections in these patients.

Existing evidence suggests that treatment with moderate-efficacy first-line DMTs (i.e., IFNβ-1b and GA) modestly increase the risk of infection, whereas high-efficacy second-line DMTs (i.e., NAT and FTY) appear to increase sharply [2]. In this regard, the observation that FTY-treated neutrophils do not display PMN intracellular killing activity in vitro may, in part, provide an explanation for the higher rates of severe infections occurring in FTY-treated patients compared to those observed in the placebo group [2,13]. Indeed, an increased susceptibility to opportunistic infections, particularly those at the level of the lower respiratory tract and skin [7], would be consistent with the immunosuppressive properties of FTY [11].

The emerging role of DMTs in increasing the risk of infections in MS patients, as attested by a growing number of studies [1,2,6,7,11,12,13,53,54], may be partially explained by the fact that MS patients treated with high-efficacy DMTs have a significantly higher disease severity compared to that of MS patients subjected to moderate-efficacy DMT treatment. Indeed, the more severe a disease, the higher the chances of contracting infections [53].

Overall, our findings further support the emerging notion that MS patients undergoing second-line DMTs are more susceptible to microbial infections due to impaired innate immunity and call for further studies aimed to deepen the role of DMTs in the modulation of other PMN functions.

In conclusion, despite being limited by the low number of DMTs tested with respect to the large amount of MS drugs available on the market, our findings warrant careful and thorough evaluation when designing personalized therapies for MS patients. In this regard, it is highly recommended that clinicians consider not only the mechanism of action of the DMT chosen but also the potential risk of infection associated with the use of such drugs in single MS patients.

## Figures and Tables

**Figure 1 jcm-09-03542-f001:**
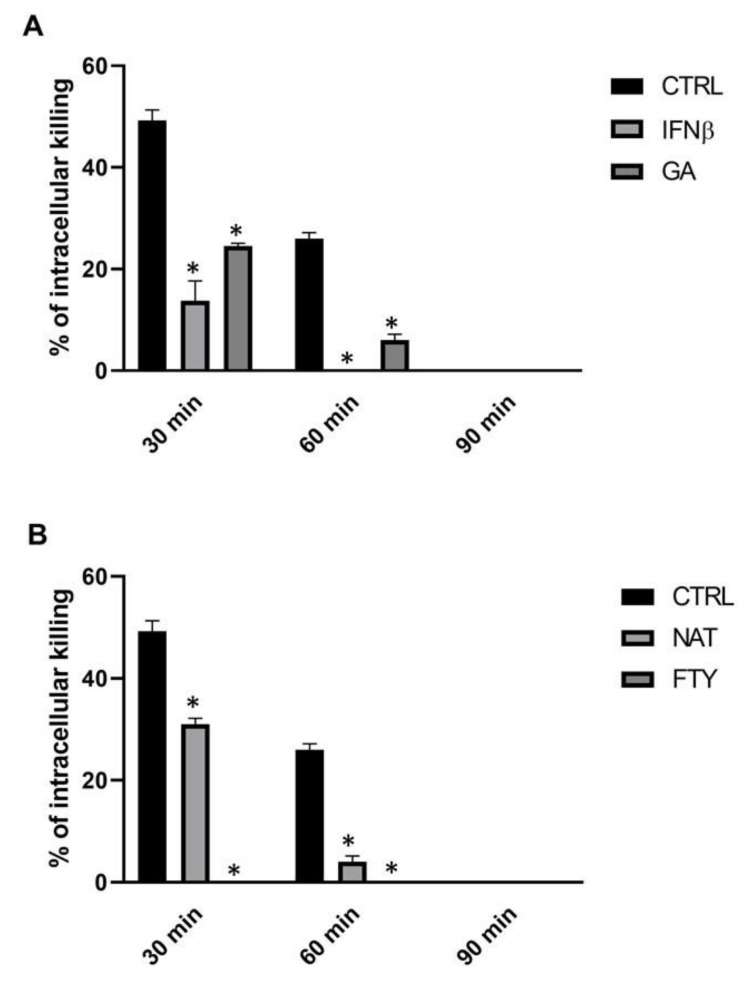
DMT treatment determined a reduction of intracellular *K. pneumoniae* killing. PMNs (10^6^ cells/mL) were incubated for 60 min with the specific DMT (**A,** first-line and **B,** second-line), then bacteria (10^7^ CFU/mL) were added, and their killing activity was evaluated at T30, T60 and T90. Data are shown as killing percentages from at least three independent experiments. * *p* < 0.01 vs. DMTs-treated cells by using the Mann–Whitney test for nonparametric data.

**Figure 2 jcm-09-03542-f002:**
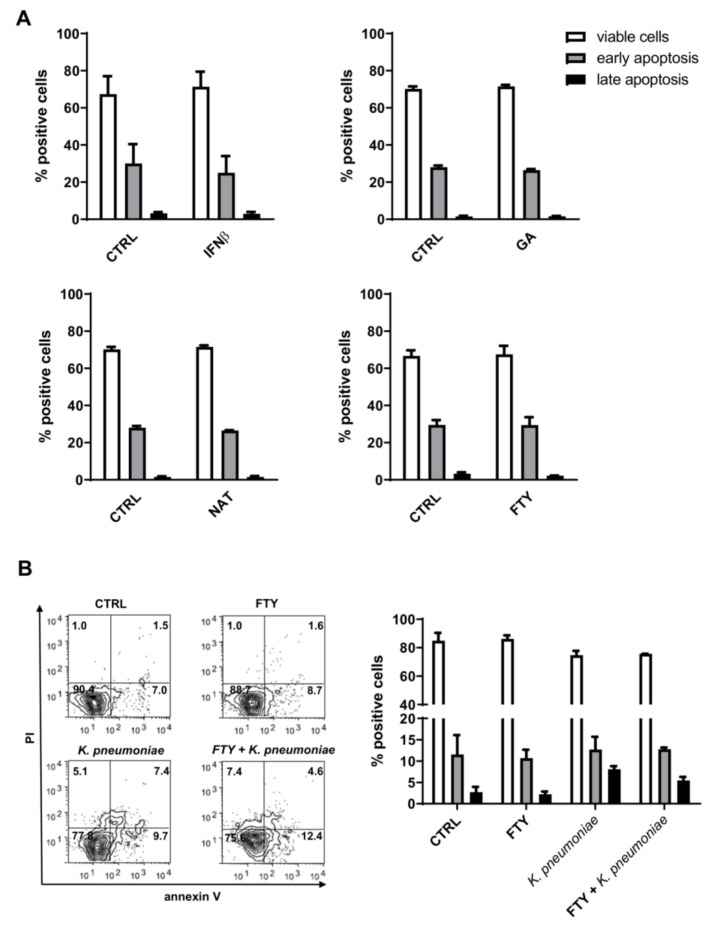
DMT treatment does not promote PMN apoptosis. Cell apoptosis was measured by annexin V-fluorescein isothiocyanate (FITC) and propidium iodide (PI) staining. (**A**) Human neutrophils were treated with the indicated DMT (IFNβ-1b, GA, NAT or FTY) for 3 h and then analyzed by flow cytometry. (**B**) Cells were pretreated with or without FTY for 1 h and then incubated with or without *K. pneumoniae* (10:1) for 2 h. Representative flow cytometry dot plots with double Annexin V-FITC/PI staining are shown in the bottom left panel. Quantitative analysis of viable neutrophils, cells in early apoptosis (annexin V positive) or in late apoptosis (annexin V positive/PI positive) are shown. Data are expressed as mean ± SEM from three independent experiments and analyzed using the Mann–Whitney test for nonparametric data.

**Figure 3 jcm-09-03542-f003:**
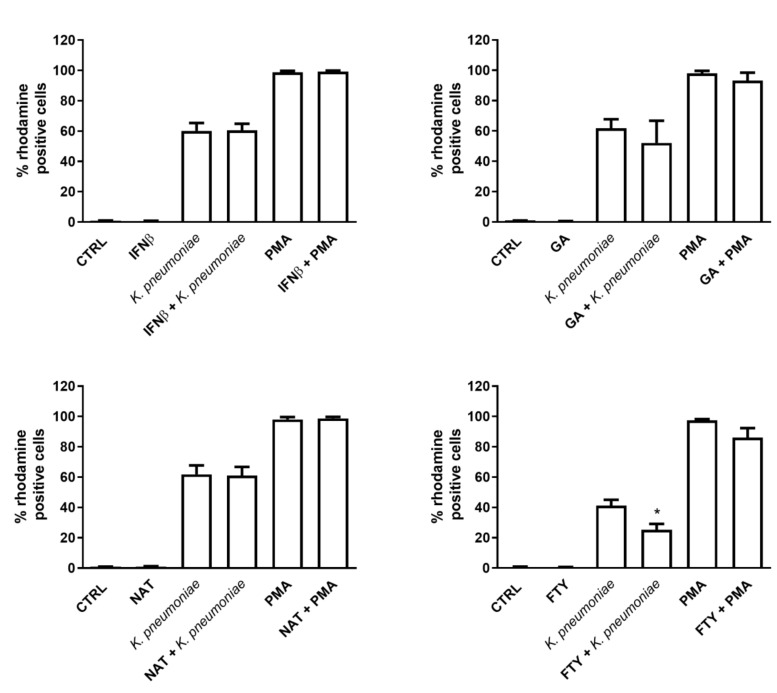
FTY exerts an inhibitory effect on intracellular ROS production. Freshly isolated human PMNs were incubated with or without the indicated DMT for 60 min at 37 °C and then exposed to *K. pneumoniae* (10:1) or 5 µM PMA for 30 min. Cells were loaded with DHR 123, and the intracellular production of ROS was assessed by FACS analysis. The percentage of PMNs expressing DHR 123 fluorescence is shown on the y axis. Data are shown as mean ± SEM from at least three independent experiments, * *p* < 0.05 as compared to the *K. pneumoniae*-stimulated sample without FTY, by using the unpaired T-test.

**Figure 4 jcm-09-03542-f004:**
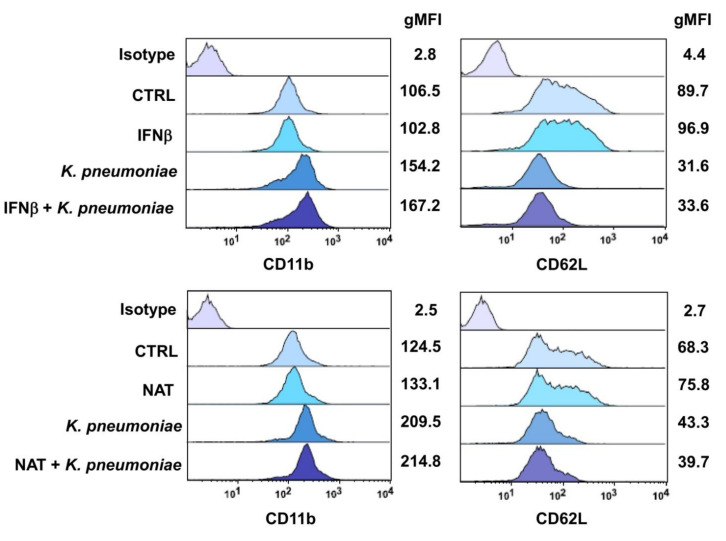
DMTs do not alter surface expression of CD11b and CD62L on resting and activated PMNs. Geometric mean fluorescence intensity (gMFI) of CD11b and CD62L on PMNs pretreated with one first-line (IFNβ-1b) or second-line (NAT) drug following *K. pneumoniae* stimulation. The four panels illustrate representative phenotypes displayed by IFNβ-PMNs (higher panels) or NAT-PMNs (lower panels), based on relative CD11b and CD62L expression levels. gMFI values relative to each marker and isotype control mAbs are shown on the right side of each panel.

**Figure 5 jcm-09-03542-f005:**
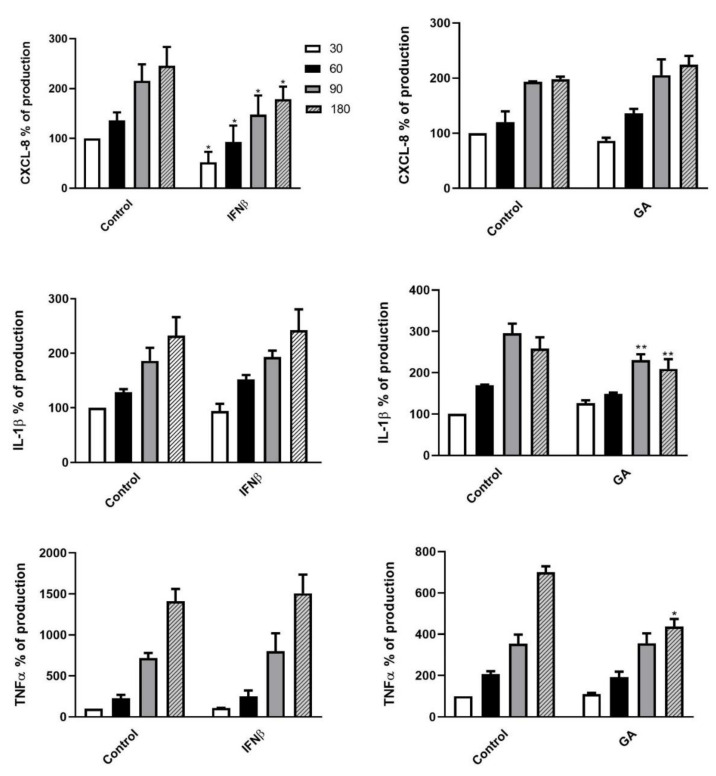
First-line DMTs reduced cytokine release by PMNs stimulated with *K. pneumoniae.* Supernatants of pretreated PMNs with IFNβ-1b (left panel) or GA (right panel), incubated in the presence of *K. pneumoniae* (ratio 10:1) for 30, 60, 90 and 180 min, were assayed to evaluate CXCL8, IL-1β and TNF-α production. Data are shown as mean ± SEM from at least three independent experiments. * *p* < 0.01 or ** *p* < 0.05 vs. untreated cells following *K. pneumoniae* challenge, by using the unpaired T-test. Results from untreated cells following *K. pneumoniae* stimulation at T30 are set as 100%.

**Figure 6 jcm-09-03542-f006:**
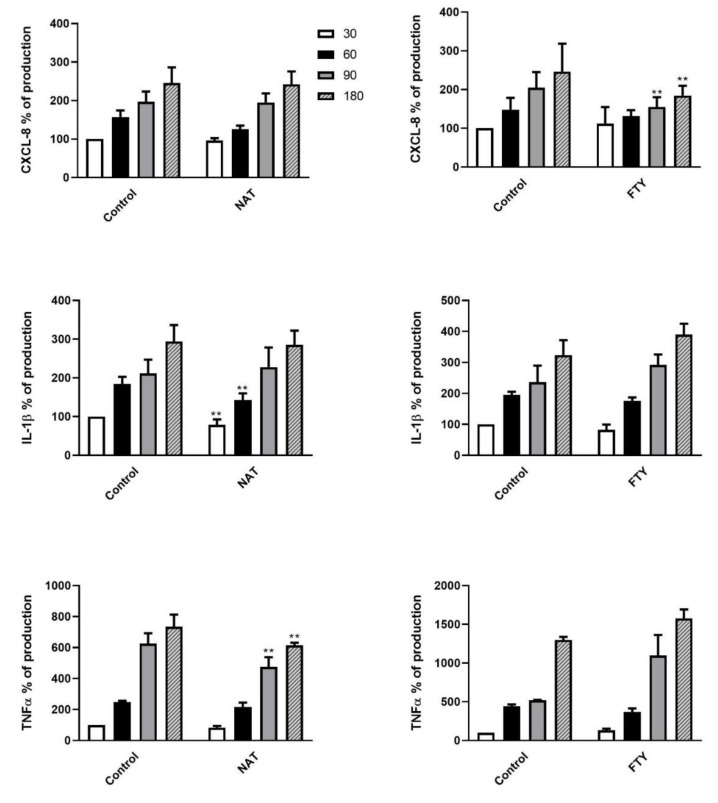
Second-line DMTs reduced cytokine release by PMNs stimulated with *K. pneumoniae.* Supernatants of pretreated PMN with NAT (left panel) or FTY (right panel), stimulated with *K. pneumoniae* (ratio 10:1) for 30, 60, 90 and 180 min, were assayed to evaluate CXCL8, IL-1β and TNF-α production. Data are shown as mean ± SEM from at least three independent experiments. ** *p* < 0.05 vs. untreated cells following *K. pneumoniae* challenge, by using the unpaired T-test. Results from unpretreated cells following *K. pneumoniae* stimulation at T30 are set as 100%.

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
