# Peer review of "Inhibition of Human Neutrophil Functions In Vitro by Multiple Sclerosis Disease-Modifying Therapies"

_jcm, 2020, doi:10.3390/jcm9113542_

Round 1

Reviewer 1 Report

Scutera et al tested in vitro the innovative hypothesis that neutrophils survival and functions are affected by drugs used to treat the multiple sclerosis patients, which could be responsible for increased risk of infections in these patients. From what I understood, the major aim is to test the viability and immune response of human neutrophils to different MS therapeutics, by measuring surface markers expression, bactericidal activity, Annexin V staining, ROS, and inflammatory cytokines. I believe science is well conducted with all appropriate controls. I have only minor issues (mostly methodological) in agreeing with the authors on their interpretations.

Issues:

  1. Please briefly discuss where appropriate, the rationale behind applied drugs concentrations (page 2, lines (69-73). Have the authors performed the dose-response analysis?
  2. On page 3 the authors describe that AnnV and PI were used to assess cell survival. Please indicate the concentrations/dilutions used (page 3, lines 98-99)
  3. Please briefly discuss where appropriate, is this description correct (page 3, lines 106-107): "FACS buffer (human PBS, 2% fetal calf 106 serum, 2 mM EDTA)"? If yes, please provide an explanation of what "human PBS" is.
  4. Please provide more details on the basis of the killing assay using Klebsiella (page 2, lines 82-86). Does this assay distinguish between the adherent bacteria from internalized bacteria (mentioned in the Discussion, lines 253-265)? In any of the cited articles, this information was not clearly explained. Please briefly discuss where appropriate, if for the killing activity, the longer time-points could be also considered or the application of Colony Forming Assay could provide more detailed information and could be considered.
  1. The authors noted no impact of tested drugs on neutrophil survival (Figure 1, lines 160-168). However, only 2-hour incubation was analyzed. Maybe a longer incubation time (16h or 24h) could reveal the difference between exposure to Klebsiella and Klebsiella combined with the drug? Along these lines, maybe authors could comment in the Discussion section on anti-apoptotic protein(s involvement in the promotion of MS in regards to their results?
  2. The authors analyzed the production of pro-inflammatory cytokines (Figure 4, lines 214-219). Please indicate where appropriate, the exact concentrations of cytokines measure [for example in pg/ml]?

Reviewer 2 Report

  1. In Material and Methods section authors claim the use of different times of incubation with DMTs for various experiments; in bacterial killing assay cells were incubated with DMTs for 1,5h, when ROS production was analyzed, incubation time was 1h whereas before apoptosis determination cells were incubated for 3h or 1h. What is the source of such discrepancies? All functions/viability should be tested after the same time of incubation with DMTs.
  2. In description of ELISA measurement authors refer to methodology that was already published, however cited article (No.16) is a review, whereas another cited article (No.17) refers to instruction of manufacturer. In such case I would suggest to write that ELISA was performed according to manufacturer’s instructions.
  3. I would suggest to change the table of bacterial killing results into graph presenting % making these results far more easy to interpret.
  4. I suggest rearrangement of results presenting apoptosis, there should be two graphs, 1 showing the data of early apoptosis (PI-/Annexin V) + as follows: ctr, IFNB, GA,NAT, FTY and the second one showing late apoptosis (PI+/Annexin V+) with same groups: ctr, IFNB, GA,NAT, FTY associated with representative dot plots with gated early and late apoptotic cells. Number of experiments performed should be included in the figures legends: “n”
  5. In the Figure 3, flow cytometry analysis reveals the % of rhodamine positive cells or MFI of rhodamine, the left axis title should be changed to make it more clear.
  6. In Figure 3, values mentioned on the right side of histograms are the mean from few experiments or they are representative values? If they are means where are SDs?
  7. ELISA is quantitative method and I strongly recommend to present data using the proper units for ex. ug/ml instead of % of control.
  8. The title of Figure 4 should be changed according to other figure legends. Title should indicate what graph is showing.

Reviewer 3 Report

In this manuscript, the authors have investigated the first line and second line DMTs prescribed to MS patients and their effects on neutrophil biology in vitro. Overall the findings suggests that DMTs such as Fingolimod, can impair anti microbial activity of Neutrophils, thus can render the patient more prone to microbial infections.

This study was well constructed and the results support all the conclusions driven. A couple of comments though,

  • Did the authors test the effect of Fingolimod on neutrophil extracellular trap formation which is also one of the most important bacterial killing mechanism of PMNs? 
  • A lot of times, second line therapies are given when benefits outweigh the risks. Authors should cite articles in the introduction section that showcases the clinical criteria used to select patients for second line therapy.

Reviewer 4 Report

In their manuscript Scutera et al. investigate the effect of DMTs used to treat MS patients on the functioning of neutrophils. During the last years it became clear that especially MS patients treated with second line DMTs are more susceptible to infections. The authors have previously shown that neutrophils from MS patients are functionally impaired and in this manuscript they compared the effect of different DMTs on bacterial killing by neutrophils, the oxidative burst, adhesion molecule expression and cytokine production. The study is largely descriptive and no mechanistical insight is provided.

Major comments

  1. The impact would be significantly improved if the authors could identify the pathway affected by the different DMTs. For instance, at time T60, the effect of the 4 DMTs is almost equal in the bacterial killing assay (26% for the control, 0% for IFNb, 6% for GA, 4% for NAT and 0% for FTY; Table 1), whereas the oxidative burst seems only to be affected by FTY. This indicates that different mechanisms are involved for FTY versus the three other drugs.
  2. For the comfort of the reader, the bacterial killing assay and the calculation of the survival index should be fully written out in the M&M.
  3. The authors focus completely on the functional response towards 1 bacterial strain, Klebsiella pneumonia. It is not clear why they specifically chose this strain. The choice should be justified. The spectrum of microorganisms should be broadened and a Gram + bacteria and a fungus should be included in the intracellular killing assay.
  4. The authors performed unpaired t-test as a statistical test, whereas all experiments were performed only 3 times. This number of independent repetitions is too low to allow statistical testing. In addition, T-tests are only appropriate when the population is normally distributed, which is impossible with n=3.
  5. The study is only observational. Why do the authors not try to reveal the pathway affected by FTY in the impairment of the respiratory burst ?
  6. In figure 3, the effect of FTY should also be shown, as it is the DMT with the greatest impact.
  7. Cytokine production was evaluated in a short time frame (Figure 4 and 5). Within 3h, the mild effect observed must be on the release of cytokines from intracellular stores rather than on de novo synthesis. If CXCL8 is affected and not IL-1beta or TNFalpha, does this mean that the cytokines are stored in different entities? Furthermore, the effect of the DMTs on cytokine release is rather limited and sometimes only observed in 2 out of the 4 time points chosen. More than half of the production remains intact, having probably only a minor biological impact. Possibly the effect would be more clear when later times would be included (cytokine production increase up to 12h after stimulation of neutropils).
  8. On line 306-308 the authors claim that NAT-induced impairment of intracellular killing is caused by a minor inhibition of cytokine release. Is there evidence that autocrine action of these cytokines is needed to exert intracellular killing activity in neutrophils ?

Minor comments

CXCL-8, should be written as CXCL8 throughout. This notation should also be adapted for the other chemokines mentioned in the discussion.

Round 2

Reviewer 4 Report

xxx